# The impact of molecular self-organisation on the atmospheric fate of a cooking aerosol proxy

Adam Milsom[1], Adam M. Squires[2], Andrew D. Ward[3] and Christian Pfrang[1,4].

[1]University of Birmingham, School of Geography, Earth and Environmental Sciences, Edgbaston, Birmingham, UK
[2]University of Bath, Department of Chemistry, South Building, Soldier Down Ln, Claverton Down, Bath, UK
[3]Central Laser Facility, STFC Rutherford Appleton Laboratory, Didcot OX11 0FA, UK
[4]Department of Meteorology, University of Reading, Whiteknights, Earley Gate, Reading, UK

*Correspondence to*: Christian Pfrang (c.pfrang@bham.ac.uk)

**Abstract.** Atmospheric aerosols influence the climate via cloud droplet nucleation and can facilitate the long-range transport of harmful pollutants. The lifetime of such aerosols can therefore determine their environmental impact. Fatty acids are found in organic aerosol emissions with oleic acid, an unsaturated fatty acid, being a large contributor to cooking emissions. As a surfactant, oleic acid can self-organise into nanostructured lamellar bilayers with its sodium salt, and this self-organisation can influence reaction kinetics. We developed a kinetic multi-layer model-based description of decay data we obtained from laboratory experiments of the ozonolysis of coated films of this self-organised system, demonstrating a decreased diffusivity for both oleic acid and ozone due to lamellar bilayer formation. Diffusivity was further inhibited by a viscous oligomer product forming in the surface layers of the film. Our results indicate that nanostructure formation can increase the reactive half-life of oleic acid by an order of days at typical indoor and outdoor atmospheric ozone concentrations. We are now able to place nanostructure formation in an atmospherically meaningful and quantifiable context. These results have implications for the transport of harmful pollutants and the climate.

## 1 Introduction

Atmospheric aerosols represent a large uncertainty when considering their impact on human-made climate change (Boucher et al., 2013) and can be associated with poor air quality in urban areas (Chan and Yao, 2008; Kulmala et al., 2021; Li et al., 2018; Molina, 2021). The organic fraction of atmospheric aerosols includes a diverse range of molecules with differing functionalities, varying with season and environment (Jimenez et al., 2009; Wang et al., 2020b).

Cooking emissions are key contributors to urban aerosols (Ots et al., 2016; Vicente et al., 2021). Fatty acids are a well-established set of marker compounds used to track cooking emissions due to their relatively high abundance (Wang et al., 2020a; Zhao et al., 2015). In particular oleic acid, an unsaturated fatty acid, has been used to follow the ageing of cooking aerosols (Wang et al., 2020a). The lifetime of oleic acid in the atmosphere is longer compared with laboratory predictions (days compared to hours) (e.g. Pfrang et al., 2011; Robinson et al., 2006; Rudich et al., 2007; Wang and Yu, 2021). This is a long-standing discrepancy and suggests that some physical process is inhibiting the ageing of such aerosols. There is also field evidence a difference in atmospheric lifetime between oleic acid and its trans isomer (elaidic acid), suggesting that the confirmation of the molecule (i.e. how the molecules organise themselves) plays a role in inhibiting reactivity (Wang and Yu, 2021). For these reasons, oleic acid has been the compound of choice for laboratory studies into aerosol heterogeneous oxidation (Gallimore et al., 2017; King et al., 2020; Milsom et al., 2021b, 2021a; Pfrang et al., 2017; Woden et al., 2021; Zahardis and Petrucci, 2007; Berkemeier et al., 2021).

The phase state and viscosity of atmospheric aerosols can impact on heterogeneous processes such as reactive gas and water uptake (Reid et al., 2018; Shiraiwa et al., 2011). Field measurements have shown that semi-solid phase formation takes place in the atmosphere (Virtanen et al., 2010) and that phase state can vary between night and day as well as with organic mass fraction (Slade et al., 2019). The long-range transport of harmful polycyclic aromatic hydrocarbons (PAHs) has been linked with particle phase state and the formation of a viscous organic layer, protecting the aerosol's potentially harmful contents (Mu et al., 2018; Shrivastava et al., 2017). Viscous phase formation is therefore a plausible explanation for the persistence of organic aerosol components in the atmosphere.

As a surfactant, oleic acid can self-organise into a range of nanostructures, known as lyotropic liquid crystals (LLCs), with its sodium salt and water (Mele et al., 2018; Seddon et al., 1990). The viscosity and the diffusion of small molecules through these phases can vary significantly (Mezzenga et al., 2005; Zabara and Mezzenga, 2014). Some nanostructures, such as the lamellar phase, have highly anisotropic diffusivities resulting in substantially higher diffusivity parallel to the lamellar bilayer compared to the perpendicular direction (Lindblom and Orädd, 1994). This nanostructure formation has been studied in levitated droplets and in coated quartz capillaries (Pfrang et al., 2017; Seddon et al., 2016), where the self-organisation of this proxy system decreased the reactivity of oleic acid by approximately an order of magnitude (Milsom et al., 2021b).

The nanostructure studied here is the lamellar phase. This consists of a bilayer of surfactant molecules with their alkyl tails directed inwards. The lamellar phase studied here is anhydrous, with no water between the bilayers (see cartoon in Fig. 1(b)). This lamellar phase is liquid crystalline, as opposed to the crystalline lamellar phase observed previously in levitated particles

(Milsom et al., 2021a). This is due to the lack of characteristic wide-angle X-ray scattering (WAXS) peaks returned by these samples (Milsom et al., 2021b), characteristic of the crystalline form of this lamellar phase (Tandon et al., 2001; Milsom et al.,

2021a). Liquid oleic acid does exhibit some order via the formation of dimers. This has been observed in the literature and we have previously confirmed this experimentally (Iwahashi et al., 1991; Milsom et al., 2021b).

In the present work, we develop a  model description of self-organised oleic acid ozonolysis and apply this both to kinetic data of the lamellar phase presented recently by Milsom *et al.* (Milsom et al., 2021b) and also to new liquid phase oleic acid ozonolysis data measured by Raman microscopy. We determine the effect on particle diffusivity of both nanostructure

formation and the formation of a later stage reaction product, which congregates in the surface region of the film. We then predict the impact on the atmospheric lifetime of such films, linking this to the discrepancy between measured and predicted atmospheric lifetimes for oleic acid (Robinson et al., 2006; Rudich et al., 2007).

## 2 Methodology

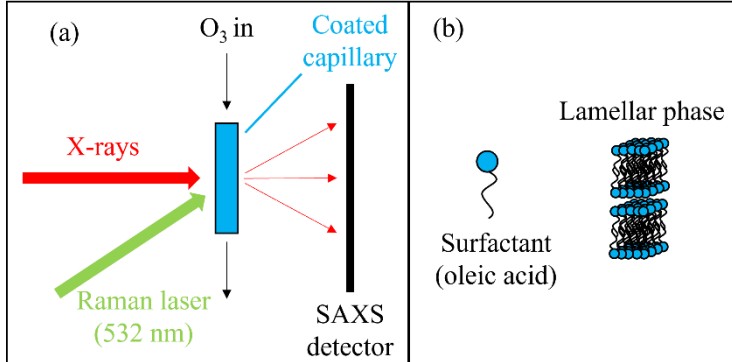

**Figure 1. (a) A schematic representation of the small-angle X-ray scattering (SAXS) and Raman spectroscopy experiments. (b) The lamellar phase formed by oleic acid and sodium oleate.**

In this study, oleic acid refers to both oleic acid and sodium oleate as they are both constituents of the lamellar phase bilayer. Pure oleic acid is referred to as liquid oleic acid. Oleic acid and sodium oleate represent the conjugate acid and conjugate base form of the same common organic aerosol component, and are expected to be present together in intermediate pH ranges (oleic

acid $pK_a$ is *ca.* 5.0).

For the present study we made liquid oleic acid capillary coatings, exposed them to ozone and followed the kinetics by Raman microscopy. The coatings were prepared by the following method (compare Milsom et al., 2021b): oleic acid (90 % purity, Sigma-Aldrich) was dissolved as a 10 wt % solution in methanol. A 70 µL aliquot of this mixture was passed up and down a quartz capillary tube (Capillary Tube Supplies Ltd., UK, 1.5 ± 0.25 mm diameter, wall thickness 0.010 mm) embedded in a

metal holder until the methanol had evaporated from the solution, aided by passing room-temperature condensed air through the capillary.

The Raman microscopy and ozonolysis experiment is based on the setup detailed in Milsom et al. (Milsom et al., 2021b). The setup is summarised along with the small-angle X-ray scattering (SAXS) experiment used to measure kinetics in the lamellar phase in Fig. 1(a). A long-working-distance objective lens (0.42 numerical aperture) was used to focus a 532-nm laser on the

capillary. The minimum spot diameter was ~1.5 µm and the laser power emitted was between 20 and 50 mW. Oxygen (BOC, 99.5%) was passed through a pen-ray ozoniser (Ultraviolet Products Ltd, Cambridge, UK) to produce ozone. The ozone concentration was calibrated offline by UV spectroscopy using the absorption cross-section for ozone at 254 nm (($1.14 \pm 0.07$) $\times 10^{-17}$ cm$^2$) (Mauersberger et al., 1986). A concentration of $77 \pm 5$ ppm was used for comparison with the lamellar phase kinetics presented by Milsom et al..

Four datasets from the same coated capillary were selected from Milsom et al. (Milsom et al., 2021b) for the following reasons: (i) they are from different sections of the same coated capillary – the experimental conditions are exactly the same ($77 \pm 5$ ppm ozone, dry oxygen-ozone flow); (ii) they have thicknesses < 5 µm – atmospherically relevant; and (iii) they are complete decays – more constraint on the model fit as the reaction was followed to completion. All error bars from these datapoints are derived from the uncertainty in the measured scattered X-ray intensity. The ozone concentration used was much higher than

that found in the atmosphere due to the major time limitations associated with synchrotron beam-time experiments.

The experimental data were modelled following the approach of the kinetic multi-layer model of aerosol surface and bulk chemistry (KM-SUB; Shiraiwa et al., 2010) based on the Pöschl-Rudich-Ammann (PRA) framework (Pöschl et al., 2007). An oleic acid ozonolysis reaction scheme was chosen where oligomer formation, viscosity and diffusivity were explicitly treated (compare Hosny et al., 2016). Our model uses a flat film geometry. The model description and the reaction scheme used are

presented in the Supplement (sect. S1).

The model was written in the *Python* programming language. A series of ordinary differential equations (ODEs) describes the change in concentration for each model component in each model bulk and surface layer over time. These ODEs were integrated using the *SciPy solve_ivp* solver with a backward differentiation formula (BDF) for stiff ODE solving (Virtanen et al., 2020).

Parameters associated with reaction rate constants, Henry's law coefficient and the gas uptake coefficient for ozone into the organic phase were set to values used in previous oleic acid ozonolysis literature (all model parameters are summarised in the Supplement, sect. S2).

The diffusion of model components throughout the film was allowed to vary with composition. It is known that self-organised phase formation affects viscosity and diffusivity (Mezzenga et al., 2005; Zabara and Mezzenga, 2014). Therefore, determining

the effect of particle diffusivity on reactivity is a key focus of this study. A Vignes-type diffusion regime was employed to account for the effect of composition on molecular diffusivity (Alpert et al., 2019; Vignes, 1966; Zhou et al., 2019a). The diffusion of model components was dependent on the fraction of lamellar oleic acid as well as dimer and trimer oligomers formed by oleic acid ozonolysis (Lee et al., 2012; Zahardis et al., 2006).

The Vignes-type diffusion parameterisation is outlined in Eq. 1 & 2.


$$D_{Y,i} = (D_{Y,lam})^{1-f_{di,i}-f_{tri,i}} \times (D_{Y,di})^{f_{di,i}} \times (D_{Y,tri})^{f_{tri,i}} \tag{1}$$

$$D_{X,i} = (D_{X,lam})^{1-f_{di,i}-f_{tri,i}} \times (D_{X,di})^{f_{di,i}} \times (D_{X,tri})^{f_{tri,i}} \tag{2}$$

*Y* in equation 1 refers to oleic acid and 9-carbon monomer products, the diffusion of which in each model layer (*i*) was treated the same under the assumption that monomer diffusion is strongly linked to oleic acid diffusion. *X* corresponds to the reactive gas, in this case ozone. The fraction of dimer ($f_{di,i}$) and trimer ($f_{tri,i}$) in model layer *i* was used to represent layer composition. The diffusion coefficients of components in the lamellar ($D_{Y,lam}$ & $D_{X,lam}$), dimer ($D_{Y,di}$ & $D_{X,di}$) and trimer ($D_{Y,tri}$ & $D_{X,tri}$) media were allowed to vary during the model fitting.

The diffusion of the dimer and trimer products was treated using power law relationship via a scaling factor ($f_{diff}$) in line with an oleic acid ozonolysis modelling study that focussed on viscosity data measurements (Hosny et al., 2016). We adapted this parameterisation to define oligomer diffusivity rather than viscosity.

$$D_{tri} = D_{di}\left(\frac{M_{di}}{M_{tri}}\right)^{f_{diff}} \tag{3}$$

$D_{tri}$ and $D_{di}$ are the diffusion coefficients of the trimer and dimer, respectively. $D_{di}$ was allowed to vary during the model fitting procedure but was not itself made to be composition-dependent. We found that the model was particularly intensive to diffusivity in the dimer (Fig. 6(b) and Fig. S3(e)). This therefore did not justify adding additional parameters and computational resource required to evolve dimer diffusivity.. $M_{di}$ and $M_{tri}$ are the respective molecular masses of the dimer and trimer products. The model output was fitted to experimental data using a differential evolution (DE) global optimisation algorithm (Storn and Price, 1997). An initial population of candidate parameter sets was created by Latin Hypercube sampling of the parameter space (McKay et al., 1979). This was carried out in parallel, similar to the procedure described by Berkemeier *et al*., who used Monte Carlo sampling to initialise their candidate parameter sets (Berkemeier et al., 2017). The DE algorithm is a popular one for finding the global minimum of a loss function used to evaluate model fitness, which in this case was the mean-squared error of the model fit. This fitting procedure was implemented using the DE method in the *optimise* module of the *SciPy* package (Virtanen et al., 2020). 20 cpu cores were used for parallelisation of the differential evolution algorithm. The model was optimised to all the datasets simultaneously, analogous to the method recently employed by Berkemeier et al. (2021). The loss function from each experimental fit was weighted according to the number of data points fitted to. Separate model optimisations to each individual dataset were carried out in order to find a range of optimised parameter values.

The sensitivity of the model to the varied parameters was investigated using an Elementary Effects algorithm via the method of Morris implementation of the *SALib Python* package (Campolongo et al., 2007; Herman and Usher, 2017; Morris, 1991). The total loss rate of oleic acid after 50 % has reacted was used as the output variable. Normalised sensitivity coefficients for each varied parameter were then obtained by measuring changes in the total loss rate of oleic acid with changes in each model parameter.

Model sensitivity to ozone solubility and surface accommodation coefficient was also explored. We found that varying the Henry's law constant by one order of magnitude more and less than that used here caused little change in the model output (Fig. S3(b)). The accommodation coefficient was also varied with some impact observed on the model output (Fig. S3(d)). Without experimental constraint on these parameters and the surface desorption lifetime of ozone, all of which are associated with surface and bulk uptake (Shiraiwa et al., 2010), a range of optimum parameter combinations is possible. We therefore decided to hold these parameters to plausible values from the modelling literature (see Table S1), highlighting the more significant effect of diffusion in this system.

We found that the model was somewhat insensitive to the branching ratio between the volatile nonanal and other monomer 9-carbon products (Fig. S3(a)). The value used in this study (0.454) agrees with experimentally determined yields (Hearn and Smith, 2004).

## 3 Results and discussion

| Film thickness / μm | Half-life / min |
|---|---|
| 0.59 (Lam.) | ~11 |
| 0.91 – 1.66 (Lam.) | ~ 18 – 22 |
| 0.6 – 0.9* (Liq.) | ~1 – 2 |

**Table 1. The half-life of the films used in this study (to the nearest minute). Taken from individual model fits to the data. *The range of half-lives from model outputs presented in Fig. 3. Lam.: Lamellar phase oleic acid; Liq.: Liquid oleic acid.**

The experimental decays, derived from SAXS peak areas, are a direct measure of oleic acid decay in the lamellar phase (Fig. 2). The half-lives of the self-organised films are significantly longer than that of liquid oleic acid (Table. 1). This suggests a significant diffusion limitation to the reaction due to the formation of this viscous self-organised phase. The half-lives of the self-organised films are also thickness dependent. Both observations are consistent with previous work on self-organised oleic acid (Milsom et al., 2021a; Milsom et al., 2021b, Pfrang et al., 2017).

It is possible to fit the model to the data without considering composition-dependent diffusion (Fig. S3(c)). The model does not fit as well to the data compared to the composition-dependent diffusion fit. In addition to this, we selected composition-dependent diffusion for a few reasons: (i) there is experimental evidence that aggregates form on the outside of levitated particles of the crystalline form of this proxy (Milsom et al. 2021a) - this aggregate is likely to be viscous; (ii) The kinetic decay from a much thicker portion of the same film (~73 μm) effectively stopped by the end of the experiment (Milsom et al., 2021b) - something, such as a viscous crust, must be stopping the reaction; (iii) The viscosity of oleic acid particles is known to increase during ozonolysis (Hosny et al., 2016). This increased viscosity is believed to affect the diffusivity of trace gases, such as ozone, through the condensed phase (Shiraiwa et al., 2011). It follows that as the composition of the film changes, so could the diffusivity of ozone.

**3.1 Diffusion behaviour**

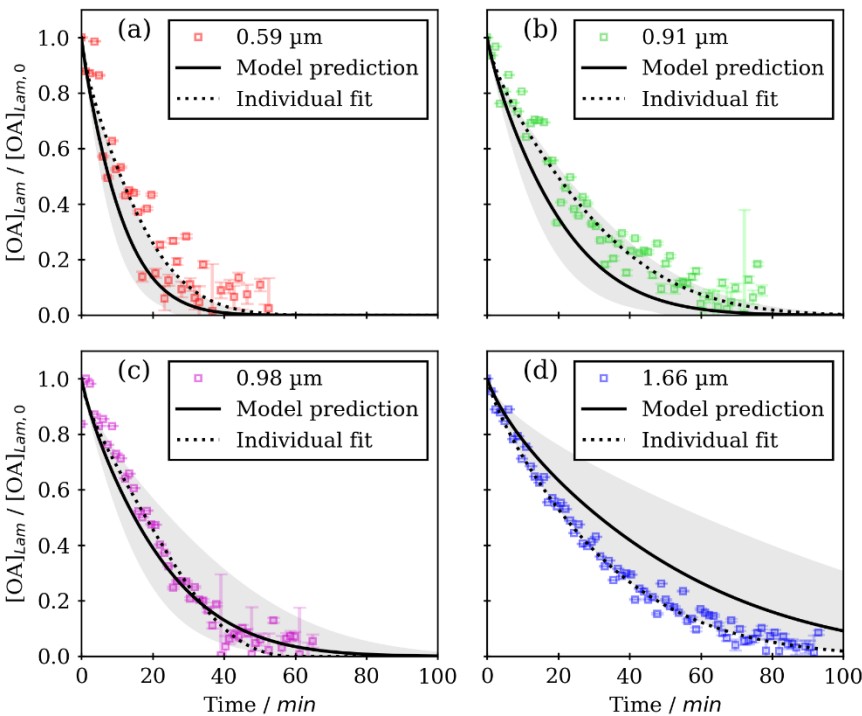

**Figure 2. Kinetic decay plots of normalised lamellar phase oleic acid concentration ($[OA]_{Lam}/[OA]_{Lam,0}$) as a function of time (experimental data from Milsom et al., 2021b); model predictions are based on the optimised model parameters determined by fitting all the data simultaneously. Individual fits to each dataset are also presented. Film thicknesses are displayed in each legend. The grey shaded regions represent the range of model outputs using parameter sets optimised from each individual fit.**

| Parameter | Description | Value*<br>/ cm² s⁻¹ | Range**<br>/ cm² s⁻¹ |
|---|---|---|---|
| $D_{dimer}$ | Bulk diffusion coefficient of the dimer | $1.03 \times 10^{-12}$ | $1.03 \times 10^{-12} - 9.49 \times 10^{-10}$ |
| $D_{trimer}$ | Bulk diffusion coefficient of the trimer | $2.07 \times 10^{-13}$ | $2.07 \times 10^{-13} - 1.90 \times 10^{-10}$ |
| $D_{X,lam}$ | Bulk diffusion coefficient of $O_3$ in the lamellar phase | $3.35 \times 10^{-12}$ | $1.13 \times 10^{-12} - 8.78 \times 10^{-8}$ |
| $D_{Y,lam}$ | Bulk diffusion coefficient of oleic acid and monomer products in the lamellar phase | $2.81 \times 10^{-12}$ | $7.32 \times 10^{-13} - 2.81 \times 10^{-12}$ |
| $D_{X,di}$ | Bulk diffusion coefficient of $O_3$ in the dimer | $4.66 \times 10^{-9}$ | $2.14 \times 10^{-9} - 7.34 \times 10^{-9}$ |
| $D_{Y,di}$ | Bulk diffusion coefficient of oleic acid and monomer products in the dimer | $8.85 \times 10^{-11}$ | $5.03 \times 10^{-11} - 9.93 \times 10^{-11}$ |
| $D_{X,tri}$ | Bulk diffusion coefficient of $O_3$ in the trimer | $1.49 \times 10^{-12}$ | $1.49 \times 10^{-12} - 9.76 \times 10^{-9}$ |
| $D_{Y,tri}$ | Bulk diffusion coefficient of oleic acid and monomer products in the trimer | $8.16 \times 10^{-11}$ | $1.24 \times 10^{-11} - 9.88 \times 10^{-11}$ |

**Table 1. Optimised diffusion parameters allowed to vary during model optimisation. The full set of model parameters is available in the Supplement (sect. S2). *From simultaneous fitting to all experimental datasets in Fig. 2. **From individual fits to each experimental dataset in Fig. 2.**

The optimised model parameters from simultaneous fitting of all datasets returned good fits for the experimental data measured at 0.59 and 0.98 µm film thickness (Fig. 2). The 0.91 µm and 1.66 µm films returned poorer fits than the other films (see succeeding discussion). A summary of the optimised diffusion parameters is presented in Table 2 and those from the individual fits are presented in Table S2 in the Supplement.

Ozone diffusion in the lamellar phase ($D_{X,lam} = 3.35 \times 10^{-12}$ cm² s⁻¹) is consistent with the diffusion of a reactive gas through a highly viscous matrix (Shiraiwa et al., 2011a). The spacing between lamellar alkyl chains in this system is 4.41 Å (Milsom et al., 2021b), which is close to the molecular diameter of ozone of 4 Å used here (Pfrang et al., 2010; Shiraiwa et al., 2010). It has been suggested that the shorter spacing between fatty acid tails on a particle surface could provide steric hindrance to diffusing ozone molecules, limiting access to the carbon-carbon double bond (Hearn et al., 2005; Vieceli et al., 2004). The anhydrous lamellar phase, being viscous and with closely-packed alkyl chains, is likely to present extra steric hindrance compared to surface monolayers because this effect would prevail throughout the film regardless of the orientation of the lamellae relative to the substrate.

Ozone diffusion in the dimer is higher than in the lamellar phase. This is consistent with a steric hindrance argument. The assumed unordered nature of the dimer product suggests that ozone can diffuse past these molecules more easily compared with diffusion through the restricted bilayers formed by the lamellar phase oleic acid. By contrast, diffusion through the trimer product is slower than in the dimer and lamellar phase. The trimer product in this model represents all the higher-order oligomers that can be formed during oleic acid ozonolysis, contributing to an increase in particle viscosity (Hosny et al., 2016).

The diffusivity of oleic acid is low in the lamellar phase ($D_{Y,lam} = 2.81 \times 10^{-12}$ cm$^2$ s$^{-1}$) compared to ~ $1.53 \times 10^{-9}$ cm$^2$ s$^{-1}$ for pure liquid oleic acid ($D_{Y,liq}$) based on its viscosity at 293.15 K (Sagdeev et al., 2019). Experimentally determined surfactant lateral diffusion coefficients in hydrated lamellar bilayers are at least four orders of magnitude higher than our model optimisation returned (~$10^{-8}$ – $10^{-6}$ cm$^2$ s$^{-1}$) (Lindblom and Orädd, 1994; Lindblom and Wennerström, 1977). Note that these experimental determinations were on hydrated lamellar phases, which are expected to be less viscous than the anhydrous lamellar phase studied here due to water acting as a plasticiser. The model does not deconvolute directionally dependent diffusion through the lamellar phase because no bilayer orientation was observed: 2-D SAXS patterns obtained for these samples did not exhibit any alignment of the lamellar phase and lamellae were randomly oriented relative to the substrate (see Fig. S2, the Supplement), though there is qualitative evidence of some degree of parallel orientation at the surface (Milsom et al., 2021b).

The diffusivity of oleic acid in the trimer product is within an order of magnitude of the lamellar phase oleic acid diffusivity. After an increase in diffusivity going from the lamellar to the dimer phase ($D_{Y,di} = 8.85 \times 10^{-11}$ cm$^2$ s$^{-1}$), oleic acid diffusivity decreases in the trimer ($D_{Y,tri} = 8.16 \times 10^{-11}$ cm$^2$ s$^{-1}$).

The diffusion constants returned from the optimised model are within the range expected for a semi-solid system (Shiraiwa et al., 2011). However, There remains significant uncertainty over the true value of some parameters (notably $D_{X,lam}$, see Table 2). Thus, we caution the over interpretation of the absolute values but we are reasonably confident in the general trends presented here.

Differences between the model and data may arise for a number of reasons associated with the experiment: (i) there is an uncertainty associated with the film thickness measurement, in particular the 0.91 and 0.98 µm films are similar when considering their quoted thickness uncertainties representing one standard deviation (0.03 µm) (Milsom et al., 2021). (ii) If the film structure changes over time exposed to ozone, which has been observed under a microscope (Hung and Tang, 2010), the surface area available for ozone uptake may also change. This change in surface structure is not considered in the model since it would require an experimental determination of the surface roughness, not possible using SAXS. (iii) The film may have been slightly thicker on one side of the capillary compared to the other: this technique required the X-ray beam to pass through both sides of a coated quartz capillary. Given that film thickness affects reaction kinetics, a difference in film thickness between both sides could impact the experimental result. (iv) The film thickness could have varied over the part of the film illuminated by the X-ray beam: the beam was ~ 320 µm × 400 µm in size and therefore the film thickness is an average of the illuminated area. These arguments could account for the range of fitted model parameters when each dataset was fitted separately (see Table 1).

We can rule out any variation in sample environment because all these datasets were obtained at different positions along the same capillary during the same ozonolysis experiment. Thus, we are confident that the film structure and morphology must have some impact on reactivity on this thickness scale.

Viscous phases have been demonstrated to drive ozonolysis chemistry down a specific route, influencing the product distribution (Zhou et al., 2019b). This is certainly possible for this viscous oleic acid system. However there does not currently

exist a product identification and distribution study for self-organised oleic acid. This is a motivation for future work in order to constrain this new aspect of the oleic acid-ozone heterogeneous system.

Following the mixing rule presented by Hosny et al. (2016), the final viscosity of these self-organised films was ~1800 mPa
s. This close to the experimental region reported for ozonised liquid oleic acid particles (~1200 – 1400 mPa s), reported as a lower estimate (Hosny et al., 2016).

In order to contrast liquid and nanostructured oleic acid kinetic decays, ozonolysis of liquid oleic coated inside a quartz capillary was carried out and followed by Raman microscopy – a technique previously used to follow oleic acid reaction kinetics (King et al., 2004; Pfrang et al., 2017). We then applied the optimised model to these experimental data, replacing the
diffusivity of ozone and oleic acid in the lamellar phase ($D_{X,lam}$ and $D_{Y,lam}$, respectively) with values for diffusion through liquid phase oleic acid ($D_{X,liq}$ and $D_{Y,liq}$, respectively). For the diffusion of ozone in liquid oleic acid we used the value from previous modelling studies on oleic acid ozonolysis (Pfrang et al., 2010; Shiraiwa et al., 2010).

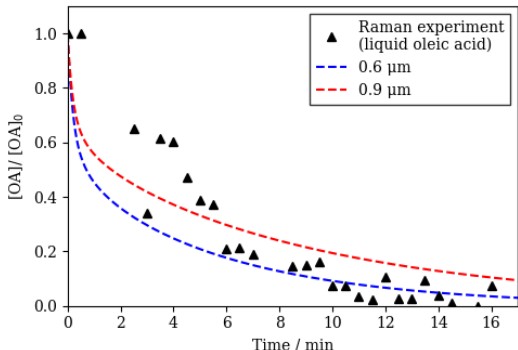

**Figure 3. Kinetic decay plot of the ozonolysis of liquid oleic acid measured by Raman microscopy. Model output for two film thicknesses (0.6 and 0.9 µm) with liquid oleic acid diffusion parameters ($D_{Y,liq} = 1.53 \times 10^{-9}$ cm$^2$ s$^{-1}$, $D_{X,liq} = 1.00 \times 10^{-5}$ cm$^2$ s$^{-1}$, replacing $D_{Y,lam}$ and $D_{X,lam}$). Experimental [O]$_3$ = 77 ± 5 ppm.**

Encouragingly, the optimised model returned a reasonable fit to ozonolysis decay data obtained by Raman spectroscopy on a film coated with pure oleic acid in the liquid state (Fig. 3). This film was prepared in the same way as the semi-solid films,
therefore it is not unreasonable to assume a similar film thickness. We varied the modelled film thickness and found that a range of film thicknesses (0.6 – 0.9 µm) fitted best to these data. Note that these data are noisier than those derived from SAXS. The concentration evolution of the model components from the fit with a 0.9 µm film thickness is presented in the Supplement (Fig. S1).


## 3.2 Spatial and temporal evolution of composition and diffusion

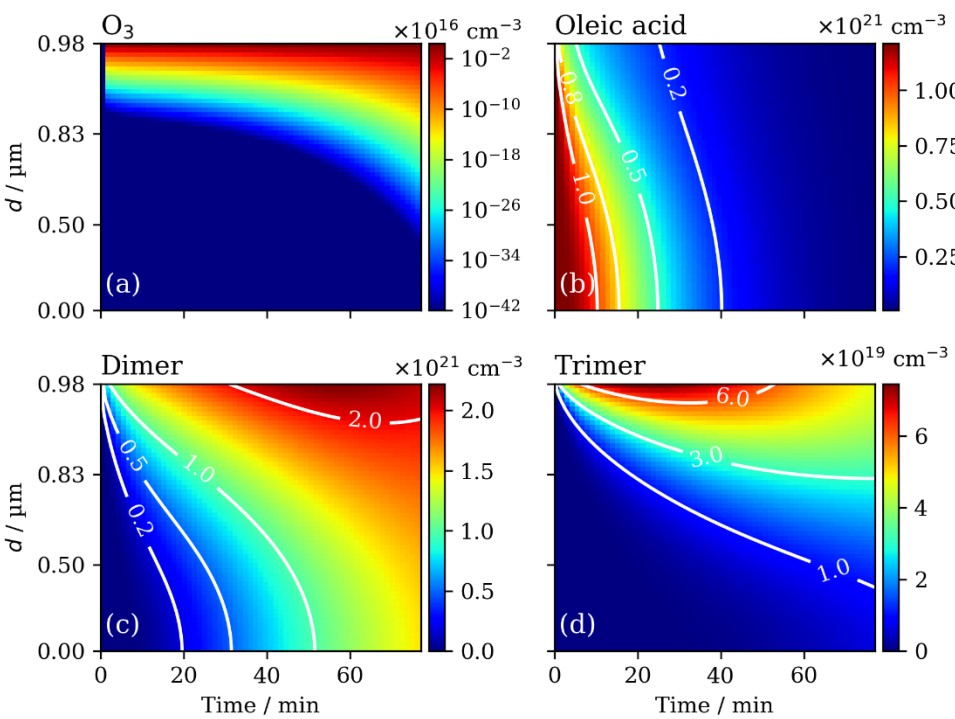

**Figure 4. Spatially and temporally resolved concentration evolution of ozone ((a) – Log-scale concentration), oleic acid (b), dimer (c) and trimer (d) model components during ozonolysis for a 0.98 μm film - d: the distance from the film-substrate interface. Contours illustrate the change in concentration gradient over time for the non-reactive gas species. [O₃] = 77 ppm.**


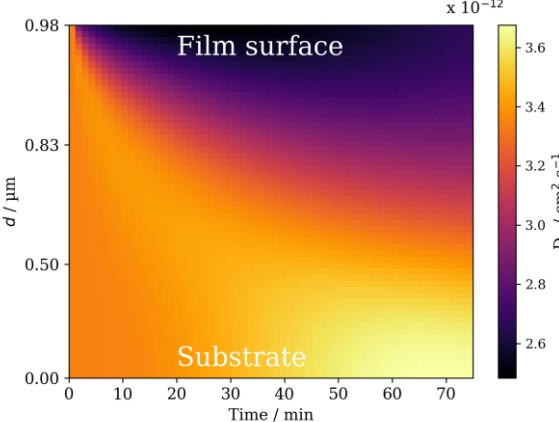

**Figure 5. The evolution of ozone diffusivity throughout a 0.98 μm film during ozonolysis. [O₃] = 77 ppm. *d*: the distance from the film-substrate interface.**

The spatial and temporal evolution of ozone concentration is consistent with a bulk diffusion-limited reaction. The concentration of ozone in the majority of the film bulk does not exceed ~ 1 % of the concentration in the surface layers (Fig. 4(a) – effectively no ozone near the film substrate). The steep ozone concentration gradient developed during the reaction is illustrated by the log-scale in Fig. 3(a).

Diffusion of ozone through regions of higher viscosity is expected to be slower and the formation of a crust in the surface
layers of the film, consisting of the viscous trimer product, inhibits the diffusion of ozone through the particle (Fig. 4(d) and Fig. 5). The formation of a surface crust has been postulated in the literature (Pfrang et al., 2011; Zhou et al., 2013) and direct experimental evidence of surface product aggregation has recently been presented in a similar proxy (Milsom et al., 2021a). Similarly, an oleic acid concentration gradient also develops during the reaction (Fig. 4(b)). This gradient is not as steep as the one observed for ozone but is still noteworthy. Surface crust formation is the source of increasing diffusive inhibition during
the reaction and therefore a key factor inhibiting the oleic acid ozonolysis kinetics for this system.

The atmospheric implications of this diffusive inhibition, caused by the initial phase state and crust formation, is explored in the *Atmospheric implications* section.

### 3.3 Kinetic regime analysis

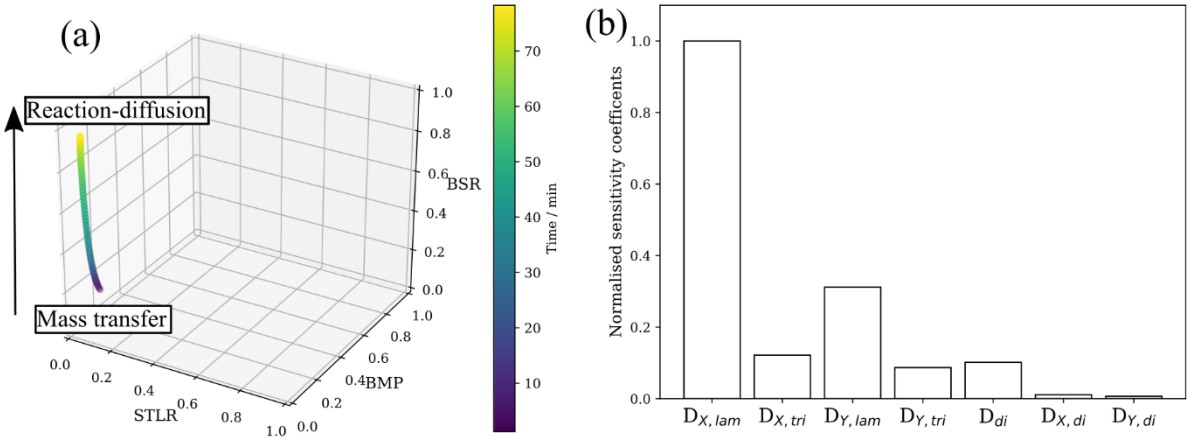


**Figure 6. (a) A "kinetic cube" plot (described by Berkemeier et al.) *(Berkemeier et al., 2013)* of surface-to-total loss ratio (STLR), bulk mixing parameter (BMP) and bulk saturation ratio (BSR) for a model run at 77 ppm ozone and 0.98 μm film thickness. The black arrow illustrates the movement from the mass transfer to the reaction-diffusion kinetic regimes described by Berkemeier et al. (b) A summary of the normalised sensitivity coefficients for each varied model parameter.**

The model output was most sensitive to ozone and oleic acid diffusivity, highlighting that film phase heavily influences its lifetime (Fig. 6(b)). From this analysis and the concentration profiles (Fig. 4) we can conclude that the reaction is limited both by bulk oleic acid diffusion to the reaction region and by the diffusivity of ozone through the film – illustrated by the

concentration gradients observed for both components (Fig. 4(a) & (b)). The model was least sensitive to diffusion coefficients of ozone and oleic acid in the dimer.

Further analysis using a method described by Berkemeier *et al.* for multi-layer model outputs demonstrates the evolution of kinetic regime as ozonolysis proceeds (Fig. 6(a)) (Berkemeier et al., 2013). The surface-to-total loss ratio (STLR) observed throughout the reaction is close to zero, suggesting that reactant loss is not a surface-dominated process. The bulk mixing parameter (BMP) starts at ~ 0.18 and decreases with time. This is a measure of how well mixed the particle is in terms of both the reactive gas and condensed phase reactant – a value of one is well-mixed. The film therefore starts poorly mixed and

becomes less well-mixed as the reaction progresses. After an initial transient phase, the bulk saturation ratio (BSR) increases steadily over time. This reflects the supply of the reactive gas to the film, which is inhibited by viscous product formation and the viscous lamellar phase.

For an appreciable amount of time the reaction regime lies between a mass-transfer and reaction-diffusion regime, illustrating the importance of both bulk diffusion and accommodation parameters at different times during the reaction (Fig. 6(a)). The

transient nature of the kinetic regime demonstrates the added insight obtained through this more explicit description. Limiting cases based on a resistor model do not account for changes in kinetic regime (Worsnop et al., 2002). This kind of analysis demonstrates the power of spatially and temporally resolved kinetic modelling, enabling us to present a more nuanced picture of the kinetic regimes underpinning this reaction.





**3.3 Atmospheric implications**

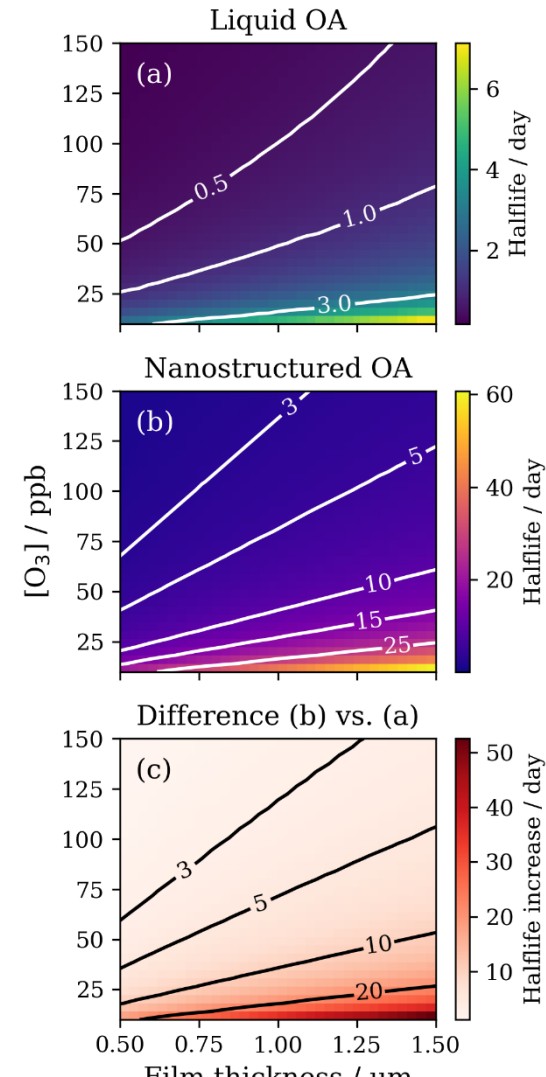

**Figure 7. Plots of film half-life as a function of ozone concentration ([O₃]) and film thickness. (a) model runs using parameters for liquid oleic acid ($D_{Y,liq} = 1.53 \times 10^{-9}$ cm² s⁻¹, $D_{X,liq} = 1.00 \times 10^{-5}$ cm² s⁻¹); (b) model runs using the optimised parameters for lamellar phase (nanostructured) oleic acid ($D_{Y,lam} = 2.81 \times 10^{-12}$ cm² s⁻¹, $D_{X,lam} = 3.35 \times 10^{-12}$ cm² s⁻¹); (c) resulting increase in half-life due to nanostructure formation. Contours on each plot represent lines of constant half-life.**

There is a known discrepancy between laboratory-determined and field-based lifetimes of fatty acids, such as oleic acid (Robinson et al., 2006; Rudich et al., 2007) and there is evidence fatty acid confirmation could affect atmospheric lifetime (Wang and Yu, 2021). In order to demonstrate the potential impact self-organisation has on the atmospheric lifetime of such organic coatings, our model was run with a film thickness range of 0.50 – 1.50 µm and an ozone concentration range of 10 –

150 ppb, covering pristine (~10 ppb), typical (20 – 40 ppb) and polluted (> 40 ppb) ozone concentrations in the urban and indoor environment (Fig. 7) (Weschler, 2000).

Taking a 1 µm film as an example, where the model agrees the best with the experiment (Fig. 2(c)), the half-life increases from ~ 1 to 10 days when moving from the liquid to the nanostructured (lamellar) state at ~ 30 ppb ozone concentration (Fig. 7(c)). Such an increase in the atmospheric lifetime of the organic film has implications for the persistence of organic matter in such particles.

These predictions are most likely an estimate of half-life, especially for the thicker films; the model over-predicts the experiment at 1.66 µm (Fig. 2(d)). Phase changes can occur with changes in relative humidity (RH) (Pfrang et al., 2017; Seddon et al., 2016). This particular system is stable below 55 % RH, above which the anhydrous lamellar phase can break down into inverse micelles which are thought to be less viscous. Atmospheric humidity is variable. Therefore, any phase transition to a less-viscous phase could enhance ozone uptake and promote a faster reaction, decreasing the half-life. The effect of different molecular arrangements are very challenging to determine experimentally (compare Milsom et al., 2021b).

An increased organic film lifetime has also direct implications for the lifetime of other particle constituents. Organic particulate matter can contain a range of chemical species, many of which are harmful to human health (Chan and Yao, 2008). The long-range transport of carcinogenic PAHs has been linked with particle phase state and the formation of a semi-solid organic coating on PAH-containing particles, increasing the risk of ill-health (Mu et al., 2018; Shrivastava et al., 2017). Our model predictions show that the semi-solidification of this atmospheric aerosol proxy can increase the lifetime of the organic film substantially. Moreover, the formation of a surface layer of high-molecular-weight products (represented as the trimer in the model) forms an extra diffusional barrier to oxidants such as ozone.

This extension of atmospheric lifetime implies a slower rate of particle oxidation. The degree of oxygenation, measured by the O:C ratio, is linked to aerosol hygroscopicity (Wu et al., 2016). Therefore, it is possible that the inhibition of particle oxidation by the formation of this semi-solid phase could have an impact on the cloud condensation nucleus (CCN) ability of the particle. The increased lifetime of oleic acid, and therefore the 9-carbon products included in this model, suggests that surface-active material can persist for longer times in the atmosphere in a semi-solid organic film. Two of the 9-carbon primary oleic acid ozonolysis products, azelaic acid and nonanoic acid, may also be surface active under certain conditions (King et al., 2009; Tuckermann, 2007, but also compare King et al., 2020). Such surface-active material has the potential to alter aerosol hygroscopicity by decreasing the surface tension of aqueous droplets, affecting the aerosol's ability to act as a CCN (Ovadnevaite et al., 2017). The link between clouds and aerosols is clear and any process affecting the ability of an aerosol to act as a CCN can have an impact on the climate (Boucher et al., 2013).

With cooking aerosols accounting for up to 10 % of reported $PM_{2.5}$ emissions in the UK (Ots et al., 2016) and fatty acids being major contributors to cooking emissions also in other regions such as China (Wang et al., 2020a), it follows that these effects would most likely be observed in the urban environment.

## 4 Conclusion

The effect of the aerosol phase state continues to be a key topic for the atmospheric aerosol community. In this study, a multi-layer kinetic model was fitted to experimental data collected during ozonolysis of oleic acid coatings in a self-organised semi-solid state.

A key advantage of this particular co-ordinated model-experiment approach is that all fitted experimental data were from samples exposed to exactly the same conditions in the same sample environment. Therefore, differences between model fits and experimental data are most likely originating from variations in film structure and morphology rather than experimental conditions, thus minimising uncertainties associated with other kinetic techniques.

The increase in atmospheric lifetime of this proxy from hours to days is consistent with field measurements of oleic acid demonstrating a much-extended atmospheric lifetime in comparison to laboratory measurements.

Future work should focus on constraining film viscosity and diffusivity experimentally and studying the effect of lamellar anisotropy on reaction kinetics. Kinetic experiments on a highly aligned lamellar phase compared with a randomly oriented lamellar phase would provide a key insight into the role a bilayer of surfactant molecules could have in hindering the uptake of trace gases to a film or particle.

We are now able to place nanostructure formation in an atmospherically meaningful and quantifiable context, thus establishing a clear pathway to determining the impact nanostructure formation could have on the atmospheric lifetime of organic aerosol emissions.

## Acknowledgements

AM acknowledges funding by the NERC SCENARIO DTP (NE/L002566/1), NERC grant (NE/T00732X/1) and support from the NERC CENTA DTP. This work was carried out with the support of the Diamond Light Source (DLS), instrument I22 (proposal SM21663). The authors are grateful to the Central Laser Facility for access to key equipment for the Raman work carried out simultaneously with the DLS beamtime experiments. Nick Terrill (DLS), Andy Smith (DLS) and Tim Snow (DLS) are acknowledged for their support during the beamtime. The computations described in this paper were performed using the University of Birmingham's BlueBEAR HPC service, which provides a High Performance Computing service to the University's research community.

## Author contributions

AM wrote the initial draft of the manuscript, wrote the model in Python and carried out the analysis and interpretation. CP contributed to the interpretation of the results and contributed to the manuscript. AMS contributed to the manuscript and discussion. ADW set up and supported the Raman microscopy experiment on I22 at DLS. All authors were involved in the Raman microscopy experiment.

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
