# Peer review of "The impact of molecular self-organisation on the atmospheric fate of a cooking aerosol proxy"

_Atmospheric Chemistry and Physics, 2021_

## Referee Comment (RC1)

Review of Milsom et al., "The impact of molecular self-organisation on the atmospheric fate of a cooking aerosol proxy"

**Overview:**

Milsom et al. describe a modelling experiment to complement previous and current laboratory work on the oleic acid/sodium oleate system, which exhibits intriguing selforganisation behaviour of interest to the atmospheric chemistry community. They demonstrate that a multi-layer kinetic model agrees well with observations of reactant loss for the oleic/oleate system and "liquid" oleic acid. They conclude that the formation of organised structures and viscous products near the film surface reduces ozone diffusion and hence significantly extends the lifetime of the double bonds in these atmospheric proxies.

Intuitively, this makes sense and is an interesting conclusion. It is qualitatively supported by the literature. I support publication in ACP but have two main concerns: Firstly, the paper dives deep quickly which limits its readability and relevance to a general ACP audience. Secondly, and more of a problem: the model treats diffusion in significant detail but I am not sure the observations they use, of oleic loss, are a sufficient constraint. How quantitatively useful the model results are is unclear. More explanation as to the limitations of the work and requirements for future observations are required.

**General comments:**

1. Self-organisation: Can the authors describe for the general reader the structure they think their oleic/oleate films have? How should "lamellar phase" be understood – crystalline vs amorphous, domains...? Is there anything special happening at the surface? Do the films show an "orientation"? Does "liquid" oleic acid exhibit any organisation? Many of these details are discussed in the authors' previous publications but it would be a good framing for the ACP community.

2. Overview of results: It would be good to see a simple discussion of the observations before the modelling discussion begins. From the first two figures, the following is apparent without modelling: the lifetime of double bonds in "liquid" oleic acid is shorter than any of the self-organised films; and the lifetime is longer for thicker films. The authors could tabulate the lifetimes (experimental fits) to make this explicit, which in turn can positively motivate the idea of diffusion limitation in general and self-organisation in particular being important. They should also discuss whether other possible explanations (e.g. differences in ozone solubility or accommodation coefficient) can be ruled out.

3. Chemical mechanism: I think the only chemical species observed/compared with here is oleic itself. If so, is the measurement-model agreement shown in Fig 1/2 sensitive to the mechanism except indirectly via diffusivity changes? The authors used a fixed chemical scheme, but how sensitive would the results be to e.g. different branching ratios?

In terms of products, I did not see diffusivity of the classic "monomer"  $C_9$  products mentioned anywhere in the main manuscript or SI. This must be rectified – how are they represented? These are the major products in previous studies (e.g. ~90% yield in the Hosny et al. study used by the authors) and would generally be expected to show an increased diffusivity compared to oleic acid or oligomers.

Finally, the authors' mechanism is based on a study of liquid oleic acid but selforganisation could drive the chemistry down a very specific route? For example, Zhou et al. (2019) observed a single secondary ozonide product in triolein ozonolysis, likely because the initial CI and carbonyl products are "caged" in a viscous matrix and so preferentially react together.

4. Diffusivity in the lamellar phase: The authors have been honest about their model performance, which is to be commended. However, some of the ranges in diffusivity shown in Table 1 are very wide, and it is not clear why. The widest range (Dx,lam, factor of 70000) corresponds to a critical parameter for oleic loss (Figure 5b). As the authors point out there are also orders of magnitude discrepancies in this parameter compared with experimental bilayer studies. This makes the quantitative value and appropriateness of the modelling hard to assess. I would like to see the authors address these wide ranges: especially, are they derived from experiment uncertainty and/or a lack of constraint on certain parameters? Further, are factor of 3 (Line 158) or 10% (Line 173) differences for diffusivities between the lamellar/dimer/trimer phases significant in this context?

This may be a motivation for future work: it suggests to me that different observations, especially of diffusivity/phase separation, are really needed. Can the viscosity of fully oxidised particles in Hosny et al. (2016) be used as a constraint? Finally, perhaps the lamellar phase needs to somehow be described differently in future, e.g. with anisotropy?

Specific comments:

Figure 1: Is it a coincidence that the modelled loss rate is higher than measured for thin films and lower than measured for thicker films?

Table 1: Please choose either "ozone" or " $O_3$ " for the descriptions.

Line 148: Why might the agreement be good for two of the films and less good for the other two? Do the authors have repeats for the experimental data or a measure of experiment uncertainty?

**References:**

Hosny et al., Chem. Sci., 1357-1367, 2016.

Zhou et al., Environ. Sci. Technol., 12467-12475, 2019.

---

## Author Comment (AC1)

**Authors' response to the reviewers of "The impact of molecular self-organisation on the atmospheric fate of a cooking aerosol proxy"**

**Referee 1**

We thank the referee for the insightful comments. We agree that a better introduction to the lamellar phase and the structure of liquid oleic acid is required in order to ease the reader into the study. A more explicit description of the model limitations and future work is also included in our revision.

**General comments:**

- 1. Self-organisation:
- (a) Can the authors describe for the general reader the structure they think their oleic/oleate films have?

A paragraph and figure (cartoon) have now been included in the methods which describe the surfactant nanostructure (lamellar phase) observed in these films. There is reference to the small-angle X-ray Scattering (SAXS) data published by our previous publication (Milsom et al. 2021b) and to the 2-D SAXS pattern in the SI (Fig. S1), which both confirm the lamellar structure and the 2-D SAXS data addresses the orientation question highlighted in point **1(c)**.

**(b) How should "lamellar phase" be understood – crystalline vs amorphous, domains...? Is there anything special happening at the surface?**

This is the liquid crystalline lamellar phase. A crystalline form of this lamellar phase has been observed in levitated particles of this proxy (Milsom et al. 2021b). However, there was no evidence in either the SAXS or Wide Angle X-ray scattering (WAXS) data for crystallinity.

From our previous study on this proxy, films of this material deposited on a silicon wafer showed a degree of orientation parallel to the silicon surface but that there is still an appreciable amount of random orientation (Milsom et al. 2021a). Therefore, nothing special is expected to happen at the surface for these samples.

These points are outlined in the extra paragraphs below.

**(c) Do the films show an "orientation"? Does "liquid" oleic acid exhibit any organisation? Many of these details are discussed in the authors' previous publications but it would be a good framing for the ACP community.**

Lamellar phase orientation is explained in our response to point **1(b)**.

Yes, liquid oleic acid does exhibit some organisation via the formation of loosely bound dimers. We showed this experimentally in our previous study (Milsom et al. 2021b) and this was previously observed in the literature (Iwahashi *et al.*, 1991). This is included in our response below.

The new paragraph in the introduction addressing points **1(a) and (c)** is as follows:

["The nanostructure studied here is the lamellar phase. This consists of a bilayer of surfactant molecules with their alkyl tails directed inwards. The lamellar phase studied here is anhydrous, with no water between the bilayers (see cartoon in Fig. 1). This lamellar phase is liquid crystalline, as opposed to the crystalline lamellar phase observed previously in levitated particles (Milsom et al., 2021a). This is due to the lack of characteristic wide-angle X-ray scattering (WAXS) peaks returned by these samples (Milsom et al., 2021b), characteristic of the crystalline form of this lamellar phase (Tandon et al., 2001; Milsom et al., 2021a). Liquid oleic acid does exhibit some order via the formation of dimers. This has been observed in the literature and we have previously confirmed this experimentally (Iwahashi et al., 1991; Milsom et al., 2021b)."]

New paragraphs in the methodology addressing point **1(c)** is as follows:

["Though the lamellar phase in these samples exhibits some degree of orientation parallel to the substrate, there is still a significant degree of random orientation (Milsom et al., 2021b). For this reason, we did not account for the effect of lamellar phase orientation on the uptake of ozone to the film."]

**2. Overview of results:**

(a) It would be good to see a simple discussion of the observations before the modelling discussion begins. From the first two figures, the following is apparent without modelling: the lifetime of double bonds in "liquid" oleic acid is shorter than any of the self-organised films; and the lifetime is longer for thicker films. The authors could tabulate the lifetimes (experimental fits) to make this explicit, which in turn can positively motivate the idea of diffusion limitation in general and self-organisation in particular being important.

We thank the referee for highlighting the lack of discussion of the experimental data. A concise description of the data is presented in our response, highlighting what exactly the decay is of (lamellar oleic acid) and referring to the already-published data showing the SAXS pattern before and after ozonolysis. We have included the idea of diffusion limitation (self-organisation) as being a key determiner of the difference between liquid and self-organised oleic acid.

We have added a new table with the half-life calculated from each experimental run, highlighting the difference between self-organised and liquid oleic acid with this paragraph at the beginning of the discussion:

["The experimental decays, derived from SAXS peak areas, are a direct measure of oleic acid decay in the lamellar phase (Fig. 2). The half-lives of the self-organised films are significantly longer than that of liquid oleic acid (Table. 1). This suggests a significant diffusion limitation to the reaction due to the formation of this viscous self-organised phase. The half-lives of the self-organised films are also thickness dependent. Both observations are consistent with previous work on self-organised oleic acid (Milsom et al., 2021a; Milsom et al., 2021b, Pfrang et al., 2017)."] New table:

| Film thickness / µm | Half-life / min |
|---------------------|-----------------|
| 0.59 (Lam.)         | ~11             |
| 0.91 – 1.66 (Lam.)  | ~ 18 – 22       |
| 0.6 – 0.9* (Liq.)   | ~1-2            |

Table 1. The half-life of the films used in this study (to the nearest minute). Taken from individual model fits to the data. \*The range of half-lives from model outputs presented in Fig. 3. Lam.: Lamellar phase oleic acid; Liq.: Liquid oleic acid.

(b) They should also discuss whether other possible explanations (e.g. differences in ozone solubility or accommodation coefficient) can be ruled out.

Generally, changes in ozone solubility have not been considered in models of oleic acid ozonolysis. We assumed that the solubility of ozone in the organic phase is unchanged between the liquid and semi-solid phases and that the uptake limitation is diffusive. To convince ourselves, we have run the model with Henry's law constants and accommodation coefficients over an order of magnitude either side of the value used in this model (new Fig. S3). Henry's law constant, desorption lifetime and the surface accommodation coefficient for ozone affect uptake into the film bulk. There is little experimental constraint on these parameters and simultaneously varying parameters associated with surface uptake/loss can return a range of "optimum" parameter sets. For these reasons, we decided not to vary these parameters.

These notions are outlined in a new paragraph in the methodology:

["Model sensitivity to ozone solubility and surface accommodation coefficient was also explored. We found that varying the Henry's law constant by one order of magnitude more and less than that used here caused little change in the model output (Fig. S3(b)). The accommodation coefficient was also varied with some impact observed on the model output (Fig. S3(d)). Without experimental constraint on these parameters and the desorption lifetime of ozone, all of which are associated with surface and bulk uptake (Shiraiwa et al., 2010), a range of optimum parameter combinations is possible. We therefore decided to hold these parameters to plausible values from the modelling literature (see Table S1), highlighting the more significant effect of diffusion in this system."]

We have removed the final paragraph in section 3.3 which repeats part of this point.

[Figure S3(b). [Part of a multi-panel plot] The effect of Henry's law constant for ozone  $(H_{cp,O3})$  on the model output for a 0.98 µm film. "Base" stands for the model output from optimisation to all datasets simultaneously (see Fig. 2, main text).]

[Figure S3(d). [Part of a multi-panel plot] The effect of the surface accommodation coefficient for ozone ( $\alpha_{s,0}$ ) on the model output for a 0.98 µm film. "Base" stands for the model output from optimisation to all datasets simultaneously (see Fig. 2, main text).]

**3. Chemical mechanism:**

(a) I think the only chemical species observed/compared with here is oleic itself. If so, is the measurement-model agreement shown in Fig 1/2 sensitive to the mechanism except indirectly via diffusivity changes? The authors used a fixed chemical scheme, but how sensitive would the results be to e.g. different branching ratios?

Not considering changes in diffusivity, changing the reaction scheme (i.e. adding specific 9-carbon products and higher-order oligomers) would not affect the decay of oleic acid as it only reacts with ozone in this model. However, the model output would be sensitive to the branching ratio because the model forms the dimer via a reaction of the non-volatile 9-carbon product with a Criegee intermediate. The branching ratio used here is what has been used previously in the literature and is consistent with experimentally determined yields (Hosny et al. 2016, Hearn and Smith, 2004).

We have run the model with different branching ratios demonstrating this sensitivity (new Fig. S3(a)) and an extra sentence has been added to the methods section referring to this branching ratio sensitivity.

The additional sentence in the methods section:

["We found that the model was somewhat insensitive to the branching ratio between the volatile nonanal and other monomer 9-carbon products (Fig. S[]). The value used in this study (0.454) agrees with experimentally determined yields (Hearn and Smith, 2004)."]

[Figure S3(a). [Part of a multi-panel plot] The effect of the branching ratio for reaction R1 (c) on the model output for a 0.98  $\mu$ m film. "Base" stands for the model output from optimisation to all datasets simultaneously (see Fig. 2, main text).]

(b) In terms of products, I did not see diffusivity of the classic "monomer" C9 products mentioned anywhere in the main manuscript or SI. This must be rectified – how are they represented? These are the major products in previous studies (e.g. ~90% yield in the Hosny et al. study used by the authors) and would generally be expected to show an increased diffusivity compared to oleic acid or oligomers.

We thank the referee for highlighting this oversight. The diffusivity of the monomer C9 products was mentioned in line 109 but is not clearly defined elsewhere. In this model, monomer diffusivity is the diffusivity of oleic acid (which is itself composition-dependent). In other words, the diffusivity of the monomer when there is a high fraction of lamellar phase oleic acid is assumed to be the same as lamellar phase oleic acid (i.e. monomers are "caged" in the viscous lamellar phase medium, which from WAXS data presented by Milsom et al. (Milsom et al., 2021b) has alkyl chain spacings on the order of the assumed diameter of these monomers {~ 0.4 nm assuming an oleic acid molecule is ~ 0.8 nm in diameter}).

We note that the diffusivity of the monomer products is not always considered explicitly in KM-SUB modelling of the oleic acid-ozone system (Shiraiwa et al., 2010; Berkemeier et al., 2021). Hosny et al. considered it as a fitted parameter - that fitted value was lower than that of oleic acid (Hosny et al., 2016), highlighting the ambiguity surrounding this parameter. We chose not to separately consider monomer diffusion for these reasons.

We have now presented this justification in the methods in the following edition to this paragraph (new sections highlighted in **bold**):

["Y in equation 1 refers to oleic acid and 9-carbon **monomer** products, the diffusion of which in each model layer (i) was treated the same **under the assumption that monomer diffusion is strongly linked to oleic acid diffusion.**"]

We have also updated the description of parameters involving  $D_Y$  in Table 2 to clarify that this is also the diffusivity of the monomer products.

(c) Finally, the authors' mechanism is based on a study of liquid oleic acid but selforganisation could drive the chemistry down a very specific route? For example, Zhou et al. (2019) observed a single secondary ozonide product in triolein ozonolysis, likely because the initial CI and carbonyl products are "caged" in a viscous matrix and so preferentially react together.

We thank the referee for bringing this study to our attention. The exact mechanism of the reaction of self-organised oleic acid is unknown and should be a subject of future work to constrain future models. It is more justifiable to go with a more well-known mechanism for the time being. This point has been added to the main text as a limitation of this study.

This paragraph has been added to the discussion, highlighting the potential impact of viscous phases on the mechanism and the need for an experiment to constrain this:

["Viscous phases have been demonstrated to drive ozonolysis chemistry down a specific route, influencing the product distribution (Zhou et al., 2019). This is certainly possible for this viscous oleic acid system. However there does not currently exist a product identification and distribution study for self-organised oleic acid. This is a motivation for future work in order to constrain this new aspect of the oleic acid-ozone heterogeneous system."]

**4. Diffusivity in the lamellar phase**

(a) The authors have been honest about their model performance, which is to be commended. However, some of the ranges in diffusivity shown in Table 1 are very wide, and it is not clear why. The widest range (Dx,lam, factor of 70000) corresponds to a critical parameter for oleic loss (Figure 5b). As the authors point out there are also orders of magnitude discrepancies in this parameter compared with experimental bilayer studies. This makes the quantitative value and appropriateness of the modelling hard to assess. I would like to see the authors address these wide ranges: especially, are they derived from experiment uncertainty and/or a lack of constraint on certain parameters?

We thank the referee for this comment. It is mainly the uncertainty in the experiment which has caused the large range of fitted values (especially for Dx,lam). We discussed these experimental limitations at length in the context of these wide ranges in the paragraph starting on line 174. To summarise that paragraph: (i) there is an uncertainty associated with the film thickness measurement; (ii) The film structure/morphology may change over time exposed to ozone (Hung and Tang, 2010) (rougher film and different surface area-volume ratio); (iii) film thickness may have been different on opposite sides of the capillary (the X-ray beam passed through both sides); (iv) film thickness could have varied in within the area illuminated by the X-ray beam.

We also recognise that surface morphology is not accounted for in this model. Attempting to do this without experimental constraint seems arbitrary but is motivation for future experimental work.

In order to highlight the context, we have added this to the paragraph starting on line 174 (changes highlighted in **bold**):

["Differences between the model and data **may** arise for a number of reasons **associated** with the experiment: (i) there is an uncertainty associated with the film thickness measurement, in particular the 0.91 and 0.98  $\mu$ m films are similar when considering their quoted thickness uncertainties representing one standard deviation (0.03  $\mu$ m) (Milsom et al., 2021b). (ii) If the film structure changes over time exposed to ozone, which has been observed under a microscope (Hung and Tang, 2010), the surface area available for ozone uptake may also change. This change in surface structure is not considered in the model since it would require an experimental determination of the surface roughness, **not possible using** SAXS. (iii) The film may have been slightly thicker on one side of the capillary compared to the other: this technique required the X-ray beam to pass through both sides of a coated quartz capillary. Given that film thickness affects reaction kinetics, a difference in film thickness between both sides could impact the experimental result. (iv) The film thickness could have varied over the part of the film illuminated by the X-ray beam: the beam was ~ 320  $\mu$ m × 400  $\mu$ m in size and therefore the film thickness is an average of the illuminated area. These arguments could account for the range of fitted model parameters when each dataset was fitted separately (see Table 1)."]

The experimental lateral diffusion coefficients presented are of surfactant diffusion (in this case, oleic acid). To the authors' knowledge there are no experimental data for the diffusion of ozone or other trace gases through the lamellar phase. We have clarified this point in line 164 onward. We also wish to correct the quoted range of experimental determinations to  $(~10^{-8} - 10^{-6} \text{ cm}^2 \text{ s}^{-1})$  on line 166. We attributed the difference between DY,lam determined here and in the literature to the anhydrous nature of our lamellar phase system. The experimental studies quoted are of the hydrated lamellar phase, with water acting as a plasticiser and reducing the viscosity of the lamellar phase. To our knowledge, there are no diffusion data associated with the anhydrous lamellar phase.

The paragraph starting from line 163 now reads as follows (changes in **bold**):

["The diffusivity of oleic acid is low in the lamellar phase ( $D_{Y,lam} = 2.81 \times 10^{-12} \text{ cm}^2 \text{ s}^{-1}$ ) compared to ~ 1.53 × 10-9 cm2 s-1 for pure liquid oleic acid ( $D_{Y,liq}$ ) based on its viscosity at 293.15 K (Sagdeev et al., 2019). Experimentally determined **surfactant** lateral diffusion coefficients in hydrated lamellar bilayers are at least four orders of magnitude higher than our model optimisation returned (~10-8 – 10-6 cm2 s-1) (Lindblom and Orädd, 1994; Lindblom and Wennerström, 1977). Note that these experimental determinations were on hydrated lamellar phases, which are expected to be less viscous than the anhydrous lamellar phase studied here due to water acting as a plasticiser. The model does not account for directionally dependent diffusion through the lamellar phase because no bilayer orientation was observed: 2-D SAXS patterns obtained for these samples did not exhibit any alignment of the lamellar phase (see Fig. S2, the Supplement)."] There is little constraint on these parameters due to the lack of available experimental data on this specific system. The values for  $D_{X,lam}$  and  $D_{Y,lam}$  are, though, in the range expected for a semi-solid system (Shiraiwa et al., 2011).

A paragraph at the end of the discussion of the fitted model values, highlighting these limitations, is added as follows:

["The diffusion constants returned from the optimised model are within the range expected for a semi-solid system (Shiraiwa et al., 2011). However, There remains significant uncertainty over the true value of some parameters (notably  $D_{x,lam}$ , see Table 2). Thus, we caution the over interpretation of the absolute values but we are reasonably confident in the general trends presented here."]

(b) Further, are factor of 3 (Line 158) or 10% (Line 173) differences for diffusivities between the lamellar/dimer/trimer phases significant in this context?

In the context of these uncertainties, we have removed these references and pointed to the general trend.

**5. Motivation for future work**

(a) It suggests to me that different observations, especially of diffusivity/phase separation, are really needed. Can the viscosity of fully oxidised particles in Hosny et al. (2016) be used as a constraint?

The issue of phase formation/separation is the subject of ongoing work.

Hosny et al. (2016) make it clear that measurements of viscosity between 0-40 % RH are lower estimates. The final viscosity for all their oleic acid particles is ~1200 - 1400 mPa s. It happens that the viscosity of our film after the reaction, using their viscosity mixing rule and viscosities derived from the diffusion coefficients of the products, is ~1800 mPa s. We carried out these experiments at close to 0 % RH (dry oxygen-ozone flow). As the values quoted by Hosny et al. are lower estimates, we are confident that we are in general agreement with that study. There is a need, however, to constrain this experimentally for self-organised oleic acid.

We have added this paragraph to the discussion highlighting this point:

["Following the mixing rule presented by Hosny et al. (2016), the final viscosity of these selforganised films was ~1800 mPa s. This close to the experimental region reported for ozonised liquid oleic acid particles (~1200 – 1400 mPa s), reported as a lower estimate (Hosny et al., 2016)."]

**(c) Finally, perhaps the lamellar phase needs to somehow be described differently in future, e.g. with anisotropy?**

Quantification of anisotropy is difficult. An ideal experiment would be to have perfectly aligned lamellar bilayers (parallel to the substrate). Reaction of these aligned bilayers with ozone, measurement of oleic acid decay and the modelling of the result could reveal the

diffusivity of ozone perpendicular to the bilayer plane. Any difference in ozone diffusivity from this value would then be attributable to anisotropy. An upcoming study by us looks at anisotropy.

Incorporating anisotropy in the model is difficult. We were able only to qualitatively show *some* degree of isotropy using Grazing-Incidence-SAXS (GI-SAXS) on these samples (Milsom et al., 2021b). A measure of the diffusion coefficient of ozone through a highly aligned lamellar phase would help us constrain this.

A paragraph has been added to the conclusion highlighting the need for more experimental constraint:

["Future work should focus on constraining film viscosity and diffusivity experimentally and studying the effect of lamellar anisotropy on reaction kinetics. Kinetic experiments on a highly aligned lamellar phase compared with a randomly oriented lamellar phase would provide a key insight into the role a bilayer of surfactant molecules could have in hindering the uptake of trace gases to a film or particle."]

**6. Specific comments**

**(a) Figure 1: Is it a coincidence that the modelled loss rate is higher than measured for thin films and lower than measured for thicker films?**

We refer to the explanations provided in response to point **4(a)**. It may be that the thicker coating is much rougher than the thinner coatings and/or there was a greater variation in film thickness in the area illuminated by the X-ray beam for this film.

**(b) Table 1: Please choose either "ozone" or "O3" for the descriptions.**

We have chosen O3.

**(c) Line 148: Why might the agreement be good for two of the films and less good for the other two? Do the authors have repeats for the experimental data or a measure of experiment uncertainty?**

We have signposted the reader to the discussion of the potential explanations for this later in this section. This is in the new paragraph updated in response to points **4(a)** and **6(a)**.

The study these data come from has a large dataset of ~50 kinetic decays to choose from (Milsom et al., 2021b). We selected these 4 datasets for a few reasons: (i) they are from different sections of the same coated capillary – the experimental conditions are the same; (ii) they have thicknesses

[Figure S3(c). [Part of a multi-panel plot] Comparison between composition-dependent and non-composition dependent diffusion. "Base" stands for the model output after optimisation to the dataset (grey squares) for a 0.98  $\mu$ m film (see Fig. 2, main text).]

5. Figure 2 – Is there a delay in reaction onset? Data does not drop until more than 1 min after the experiments starts, but then drops very quickly. Also, the film thickness seems to be used as fitting parameter. Is there a way of verifying these 0.9  $\mu$ m? Reporting a possible range (e.g. with a shading) might overcome a potential arbitrariness of the fit parameter.

We thank the referee for pointing this out. There was a mistake in measuring the relevant peak areas in the Raman spectra. The original data were normalised to peaks with a large background due to some lights that were switched on in the experiment room part way through ozonolysis. Those datapoints are now omitted. Re-integration of those peaks and normalising to the correct "time = 0 s" spectrum has returned a more characteristic kinetic decay (see new Fig. 3). The optimised fit to these data using the film thickness as a fitting parameter now shows the film thickness as 0.8  $\mu$ m. We also note that kinetic data derived from Raman spectroscopy are much noisier than those derived from SAXS.

We have plotted a range of model outputs with different film thicknesses as suggested (see new Fig. 3).

[Figure 3. Kinetic decay plot of the ozonolysis of liquid oleic acid measured by Raman microscopy. Model output for two film thicknesses (0.6 and 0.9  $\mu$ m) with liquid oleic acid diffusion parameters ( $D_{Y,liq} = 1.53 \times 10^{-9} \text{ cm}^2 \text{ s}^{-1}$ ,  $D_{X,liq} = 1.00 \times 10^{-5} \text{ cm}^2 \text{ s}^{-1}$ , replacing  $D_{Y,lam}$  and  $D_{X,lam}$ ). Experimental [ $O_3$ ] = 77 ± 5 ppm.]

We have updated the paragraph discussing this figure (changes in **bold**):

[Encouragingly, the optimised model returned a **reasonable** fit to ozonolysis decay data obtained by Raman spectroscopy on a film coated with pure oleic acid in the liquid state (Fig. **3**). This film was prepared in the same way as the semi-solid films, therefore it is not unreasonable to assume a similar film thickness. We varied the modelled film thickness and found that **a range of film thicknesses (0.6 – 0.9 µm)** fitted best to these data. **Note that these data are noisier than those derived from SAXS**. The concentration evolution of the model components from **the fit with a 0.9 µm film thickness** is presented in the Supplement (Fig. S1).] 6. Is line 154 ("the shorter spacing between fatty acid tails on a particle surface could provide steric hindrance to diffusing ozone molecules, limiting access to the double bond") at odds with line 169 ("2-D SAXS patterns obtained for these samples did not exhibit any alignment of the lamellar phase")? Would one expect a difference in diffusion between "entry" of the layer and diffusion within the layer? Instead of correcting the bulk diffusion coefficient, could or should the bulk accommodation coefficient be fitted?

We understand the confusion arising from these two sentences. Line 154 states what has been suggested in the literature. The sentence after the one in line 154 goes on to explain that this effect would prevail throughout the bulk of the lamellar phase due to the effect of closely-packed alkyl chains.

We have updated this sentence in the main text to clarify (changes in **bold**):

["The anhydrous lamellar phase, being viscous and **with closely-packed alkyl chains**, is likely to present extra steric hindrance compared to surface monolayers because this effect would prevail throughout the film **regardless of the orientation of the lamellae relative to the substrate**."]

Alignment refers to the degree of parallel and random orientation of the lamellar phase relative to the substrate. The alkyl chains are still closely packed in the lamellar phase no matter the orientation relative to the substrate, supporting the steric hinderance argument. One would expect a difference between "entry" and diffusion within the layer if there was a significant amount of parallel alignment relative to the substrate. This can be qualitatively measured using GI-SAXS. Though there was some evidence of this at the surface (using GI-SAXS – Milsom et al., 2021b), both the GI-SAXS and 2-D SAXS patterns return significant random orientation of the lamellar phase (Fig. S2).

We have updated the sentence in line 169 to clarify this:

["The model does not **deconvolute** directionally dependent diffusion through the lamellar phase because no bilayer orientation was observed: 2-D SAXS patterns obtained for these samples did not exhibit any alignment of the lamellar phase **and lamellae were randomly oriented relative to the substrate** (see Fig. S2, the Supplement), **though there is qualitative evidence of some degree of parallel orientation at the surface (Milsom et al., 2021b)**."]

We refer to our response to point **2(b)** to referee 1 for an explanation of why we did not vary the accommodation coefficient.

**7. Minor and technical comments**

(a) I. 46 – There seems to be a word missing in or after "has been linked with particle phase".

We have updated this sentence as follows (change in **bold**):

["The long-range transport of harmful polycyclic aromatic hydrocarbons (PAHs) has been linked with particle phase **state** and the formation of a viscous organic layer..."]

**(b) I. 56 – It is difficult to wrap one's head around "self-organised phase resolved model description" and what it may mean.**

We have updated this sentence as follows (change in **bold**):

["In the present work, we develop model description of **self-organised** oleic acid ozonolysis..."]

(c) I. 82 – "The selected decays were from the same experiment carried out simultaneously in the same capillary under the same conditions" – It is not clear what this means.

See our response to point **6(c)** for referee 1, which clarifies what we mean by this and replaces most of that paragraph.

(d) I. 133 - Fixing the model time at which one looks for sensitivity tests can have pitfalls. A quicker initial oleic acid decay could lead here to depletion before 40 minutes and hence a lowering of the loss rate at 40 minutes. I would suggest comparing loss rates at similar reaction coordinate / progress (e.g. loss rate after 50 % has reacted).

We thank the referee for this insight. The sensitivity analysis has been carried out again using the loss rate after 50 % of the oleic acid had reacted. We have updated Fig. 6 and amended the text accordingly:

---

## Author Comment (AC2)

**Supporting information for "The impact of molecular self-organisation on the atmospheric fate of a cooking aerosol proxy".**

Adam Milsom[1], Adam M. Squires[2], Andrew D. Ward[3] and Christian Pfrang[1,4].

[1]University of Birmingham, School of Geography, Earth and Environmental Sciences, Edgbaston, Birmingham, UK
[2]University of Bath, Department of Chemistry, South Building, Soldier Down Ln, Claverton Down, Bath, UK
[3]Central Laser Facility, STFC Rutherford Appleton Laboratory, Didcot OX11 0FA, UK
[4]Department of Meteorology, University of Reading, Whiteknights, Earley Gate, Reading, UK

**S1. Model reaction scheme**

(R1) $O_3$ + oleic acid → c(NN + CI) + (1-c)($C_9$ + CI)

(R2) CI + C9 → dimer

(R3) CI + dimer → trimer

A simplified form of the reaction scheme used by Hosny *et al*. has been used (Hosny et al., 2016). The initial reaction of oleic acid with ozone ($O_3$) forms nonanal (NN) or a 9-carbon ($C_9$) product (nonanoic acid, 9-oxononanoic acid or azelaic acid) plus a Criegee intermediate (CI). The relative amount of NN and $C_9$ products is determined by the branching ratio (c). We have also included the loss of NN from the surface due to its high volatility. The CI then goes onto react with a $C_9$ product to form the dimer. This dimer then reacts with another CI to form the trimer, which represents all higher order products in this model scheme.

Since this is a simplified reaction scheme and we assume that diffusion parameters dominate the reaction kinetics, we have held all reaction rates to those optimised by Hosny et al. (Hosny et al., 2016) (see next section).

**S2. Model parameters**

[revised manuscript text omitted]